# Evolutionary origin of genomic structural variations in domestic yaks

Xinfeng Liu[1,2,3,7], Wenyu Liu[1,7], Johannes A. Lenstra [4,7], Zeyu Zheng [1,7], Xiaoyun Wu[2,7], Jiao Yang[1], Bowen Li[1], Yongzhi Yang[1], Qiang Qiu [1], Hongyu Liu[5], Kexin Li[1], Chunnian Liang[2], Xian Guo[2], Xiaoming Ma[2], Richard J. Abbott[6], Minghui Kang [1] ✉, Ping Yan[2] ✉ & Jianquan Liu [1,3] ✉

Yak has been subject to natural selection, human domestication and interspecific introgression during its evolution. However, genetic variants favored by each of these processes have not been distinguished previously. We constructed a graph-genome for 47 genomes of 7 cross-fertile bovine species. This allowed detection of 57,432 high-resolution structural variants (SVs) within and across the species, which were genotyped in 386 individuals. We distinguished the evolutionary origins of diverse SVs in domestic yaks by phylogenetic analyses. We further identified 334 genes overlapping with SVs in domestic yaks that bore potential signals of selection from wild yaks, plus an additional 686 genes introgressed from cattle. Nearly 90% of the domestic yaks were introgressed by cattle. Introgression of an SV spanning the *KIT* gene triggered the breeding of white domestic yaks. We validated a significant association of the selected stratified SVs with gene expression, which contributes to phenotypic variations. Our results highlight that SVs of different origins contribute to the phenotypic diversity of domestic yaks.

Yaks are members of the tribe *Bovini*, which comprises several other large domestic or wild ruminant species exhibiting different degrees of cross-fertility and contrasting climate adaptations[1–4]. The wild yak (*Bos mutus*), of which there are now around 15,000 individuals in the wild, has from ~5 million years ago been adapted to conditions on the Qinghai-Tibet Plateau (QTP), where temperatures and oxygen concentrations are extremely low at elevations between 4000 and 6000 m[1,2,4–6]. The domestic yak (*Bos grunniens*), now numbering around 18 million individuals, is normally kept between 3000 and 5000 m, providing food and other commodities[1,2,4–6]. The introduction of cattle to the QTP led to hybridization and introgression with domestic yaks[4]. Thus, the

domestic yak has been subject to natural selection, artificial selection (during domestication) and interspecific introgression with cattle during its evolution. It is, therefore, a useful research model for studying genetic changes resulting from these evolutionary processes.

Long-read sequencing of a limited number of yaks has revealed that genomic structural variants (SVs) have been important in both high-altitude acclimation and domestication of yaks[7,8]. The construction of graph-based pangenomes of species and super-pangenomes across species allows a comprehensive identification of evolutionary origins of SVs, which substantially modulate gene expression and partially explain the hidden heritability and genetic introgression from

[1]State Key Laboratory of Herbage Improvement and Grassland Agro-ecosystem, College of Ecology, Lanzhou University, Lanzhou 730000, China. [2]Key Laboratory of Animal Genetics and Breeding on Tibetan Plateau, Ministry of Agriculture and Rural Affairs, Lanzhou Institute of Husbandry and Pharmaceutical Sciences, Chinese Academy of Agricultural Sciences, Lanzhou 730050, China. [3]Academy of Plateau Science and Sustainability, Qinghai Normal University, Xining 810016, China. [4]Faculty of Veterinary Medicine, Utrecht University, Utrecht 3508 TD, The Netherlands. [5]Anhui Provincial Laboratory of Local Livestock and Poultry Genetical Resource Conservation and Breeding, College of Animal Science and Technology, Anhui Agricultural University, Hefei 230036, China. [6]School of Biology, University of St Andrews, St Andrews KY16 9AJ, UK. [7]These authors contributed equally: Xinfeng Liu, Wenyu Liu, Johannes A. Lenstra, Zeyu Zheng, Xiaoyun Wu. ✉e-mail: kangminghui0106@gmail.com; yanping@caas.cn; liujq@lzu.edu.cn

the closely related species[9–17]. However, our understanding of the evolutionary origin and functional role of such SVs in domestic yaks is restricted by the low quality and limited number of reported whole-genome sequences (WGS) available for yaks and their close relatives. To overcome this, we report here a pangenomic analysis of 28 de novo WGS assemblies for wild and domestic yaks, and Asian cattle, plus 19 published assemblies of yak, cattle, bison, wisent and gaur[9,10,12,18–24]. SVs were genotyped in a larger panel of 386 mainly short-read resequenced individuals from 7 species and were diagnosed as contributing to hypoxic adaptation, yak domestication and adaptive introgression from cattle. These findings considerably increase our understanding of genome diversity and the genetic basis of adaptation in wild and domestic yaks.

## Results

### Phylogenomic relationships and a pangenome of wild and domestic yaks

To characterize genome diversity in both wild and domestic yaks, de novo WGS assemblies were constructed for 6 wild and 15 domestic yaks representing their full distribution range, plus two low-altitude Asian zebu and four high-altitude taurine cattle, and one high-altitude taurine-zebu-yak hybrid (Zhangmu cattle, ZMC) (Table 1, Supplementary Figs. 1, 2a, Supplementary Data 2, 3 and 4). These data were combined with previously reported genomes for one wild yak, 14 cattle, two bison, one wisent and one gaur, resulting in a panel of genomic datasets of the *Bovini* (Table 1, Supplementary Figs. 1, 2a, Supplementary Data 2, 3 and 4)[9,10,12,18–24]. For consistent analysis, we used a uniform standard pipeline to annotate these 47 bovine genomes (Table 1, Supplementary Data 5). We identified an average of 24,368 protein-

coding genes for each assembly (Table 1, Supplementary Data 5), and BUSCO evaluation of the protein-coding annotations showed that an average of 97. 4% (Supplementary Data 5) were completely annotated, indicating the high completeness of the gene annotation.

Across these genome assemblies 1,048,639 high-confidence SNPs were detected and used in phylogenetic analyses with the water buffalo genome as outgroup (Fig. 1a). We further constructed a species tree on the basis of 8428 single-copy core genes through selecting one representative individual of each species (Fig. 1b). Both trees agree well with previous studies[3,4], although domestic yaks did not form a monophyletic cluster (Fig. 1a).

Pangenomes were constructed for yaks and cattle and a super-pangenome for the 7 *Bovini* species (Fig. 1c, Supplementary Fig. 2b–h). For the yak pangenome the total gene set approached saturation at $n = 20$ (Fig. 1c). The percentages of core (present in all 22 genomes), near-core (present in 20–21 genomes) and variable (found in 1–19 genomes) gene families were 50.18, 10.91, 38.91%, respectively (Fig. 1d), with core and near-core genes showing higher mean levels of gene expression and lower mean *Ka/Ks* ratios (Fig. 1e, f). A pangenome for cattle and a super-pangenome for 47 genomes of 7 *Bovini* species showed similar trends (Supplementary Fig. 2b–h). On average, the domestic yak assemblies contained 119 genes that were not present in the wild yak assemblies, 56 of which originated from cattle by introgression (Fig. 1g) and were enriched for disease defence, development and reproduction (Supplementary Fig. 10a). In pairwise comparisons of the assemblies constituting the pangenome, each assembly possessed 123 to 2113 genes not present in the other genome (Fig. 1h).

**Table 1 | Assembly and annotation statistics for the 28 novel genomes generated in this study**

| Sample ID | Subpopulation | Contig N50 (Mb) | Assembly Size (Gb) | Repetitive Content (%) | Quality value | Gene number |
|---|---|---|---|---|---|---|
| DYGS03 | *B. grunniens* | 3.64 | 2.63 | 42 | 38.4 | 23,668 |
| DYGS37 | *B. grunniens* | 3.30 | 2.66 | 42 | 39.5 | 24,224 |
| DYNP09 | *B. grunniens* | 60.72 | 2.80 | 58 | 41.1 | 23,968 |
| DYPK16 | *B. grunniens* | 51.57 | 2.74 | 57 | 40.9 | 23,368 |
| DYQH01 | *B. grunniens* | 1.02 | 2.65 | 42 | 37.9 | 23,278 |
| DYQH13 | *B. grunniens* | 32.50 | 2.74 | 43 | 39.9 | 24,586 |
| DYSC21 | *B. grunniens* | 22.70 | 2.78 | 44 | 37.4 | 23,616 |
| DYSC29 | *B. grunniens* | 28.51 | 2.70 | 56 | 40.0 | 23,364 |
| DYXJ17 | *B. grunniens* | 35.63 | 2.74 | 43 | 38.3 | 24,138 |
| DYXZ30 | *B. grunniens* | 69.40 | 2.75 | 55 | 39.3 | 23,812 |
| DYXZ38 | *B. grunniens* | 28.95 | 2.81 | 45 | 40.0 | 24,656 |
| DYXZ91 | *B. grunniens* | 0.85 | 2.61 | 42 | 38.3 | 23,738 |
| DYXZ92 | *B. grunniens* | 73.03 | 2.81 | 60 | 68.3 | 23,712 |
| DYYN03 | *B. grunniens* | 40.36 | 2.69 | 43 | 40.7 | 24,336 |
| DYYN14 | *B. grunniens* | 56.41 | 2.74 | 43 | 41.8 | 23,906 |
| WY11 | *B. mutus* | 103.53 | 2.93 | 52 | 60.0 | 25,006 |
| WY12 | *B. mutus* | 60.75 | 2.75 | 44 | 40.7 | 23,776 |
| WY15 | *B. mutus* | 13.97 | 2.72 | 43 | 40.0 | 25,018 |
| WY16 | *B. mutus* | 94.41 | 3.03 | 63 | 56.6 | 25,002 |
| WY17 | *B. mutus* | 22.06 | 2.74 | 43 | 40.2 | 25,040 |
| WY18 | *B. mutus* | 65.03 | 2.97 | 57 | 62.1 | 25,093 |
| CDMC01 | *B. taurus* | 8.48 | 2.72 | 43 | 41.4 | 23,760 |
| TC13 | *B. taurus* | 18.60 | 2.89 | 62 | 40.0 | 24,374 |
| TC14 | *B. taurus* | 13.62 | 2.86 | 60 | 39.0 | 23,866 |
| TC21 | *B. taurus* | 2.53 | 2.65 | 42 | 23.0 | 24,054 |
| WNC04 | *B. indicus* | 75.04 | 2.81 | 60 | 64.8 | 22,272 |
| Brahman01 | *B. indicus* | 66.70 | 2.84 | 60 | 66.5 | 25,344 |
| ZMC01 | *B. taurus × indicus×grunniens* | 1.92 | 2.69 | 42 | 33.3 | 23,372 |

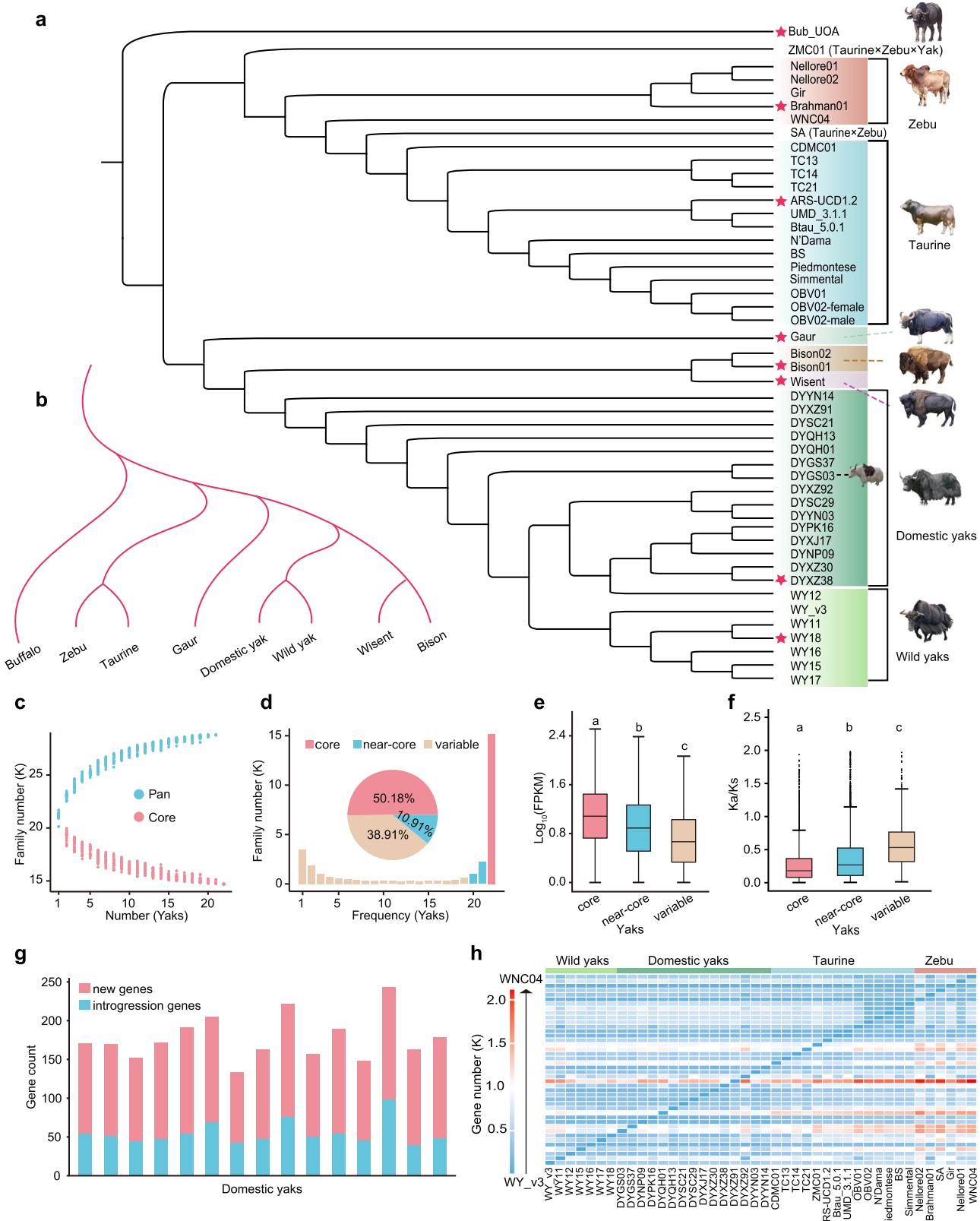

## Bovine graph-genome and characterization of structural variants (SVs) across 386 individuals from 7 species

For identifying SVs, we constructed a multi-assembly graph-based genome of the 47 genomes used in the phylogenomic and super-pangenome analyses. This comprised 3.14 gigabases (Gb) spread across 5,449,222 nodes (the number of fragments of sequences) and connected by 4,889,530 edges (the connections between nodes), with non-reference nodes spanning 387.0 Mb (Fig. 2a, b). The core (shared by all genomes), near-core (in 46 or 45 genomes) and variable nodes (in 44 or less samples) accounted for 60.8, 17.0, 22.2% of all nodes (Fig. 2b, Supplementary Data 6).

We detected SVs (≥ 50 bp) in the graph-based genome using the bubble popping algorithm of gfatools[25] and retained 293,712 SVs (81.7% <500 bp, 99.76% <10 kb) that could be genotyped in the

**Fig. 1 | Phylogeny of bovine species and pan-genome of wild and domestic yaks.**
**a** Phylogenetic tree of 47 bovine individuals with de novo genomes-based SNPs
with buffalo as outgroup. CDMC Chaidamu cattle, TC Tibetan cattle, ZMC Zhangmu
cattle, WNC Wannan cattle, BS Brown Swiss, OBV Original Braunvieh, SA Sanga
Ankole, ARS-UCD1.2, Btau_5.0.1 and UMD_3.1.1: Hereford cattle. **b** A cladogram of
*Bovini* species based on single-copy genes in the individuals marked with a red star
in Fig. 1a. **c** Variation of gene families in the pan-genome and core genome along
with an additional yak genome. **d** Numbers and frequencies of core, near-core, and
variable gene families. **e**, **f** The gene expression levels and *Ka/Ks* values of core

(16,900; 12,532), near-core (4186; 2875), and variable genes (2934; 774) of yaks.
Lowercase letters indicate significant differences ($p < 0.05$); two-sided Student's *t*-
test. Box edges indicate the upper and lower quartiles, the centerlines indicate the
median value, the horizontal bars indicate the maximum and minimum values, and
the dots indicate outliers. **g** numbers of genes without homologues in the wild
assemblies ("new genes") and genes originating from cattle as inferred from their
phylogenetic position ("introgression genes"). **h** Heat map representing the num-
ber of genes present in the assembly on the X-axis but not in the assembly on the
*Y*-axis.

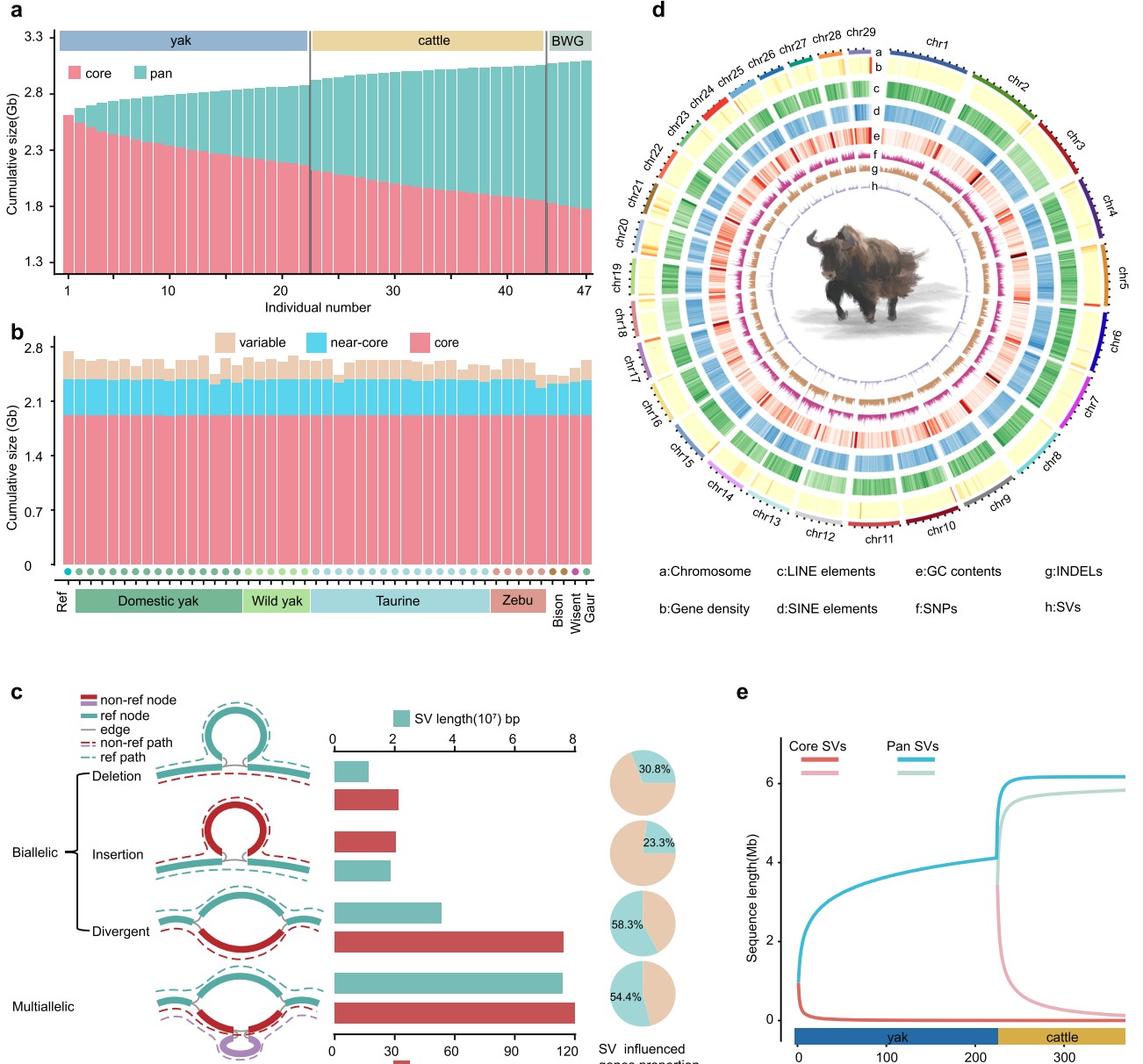

**Fig. 2 | Characterization of the graph-genome of 47 de novo bovine genomes
and distributions of structural variants (SVs) across these genomes and 386
individuals. a** Decrease and increase of the core nodes (red) and pan nodes (red +
blue) respectively with the increase in number of yak, cattle and bison/wisent/gaur
(BWG) genome assemblies added to the graph-genome. **b** The number of nodes in
each discovery class (core, near-core, and variable) is shown per assembly.
**c** Diagrams of diverse SV types from bovine graph pangenome based on the yak

reference genome BosMut3.0. The bar chart shows the average number and length
of each type of SV separately. The pie chart shows the number of genes affected by
SV as a proportion of the overall number of genes. **d** Genomic features and varia-
tion landscape across the reference genome assembly. SNPs, INDELs, and SVs from
resequencing data. **e** Pan-SV and core-SV length with 386 individuals of 7 bovine
species. The light pink and green lines at the right indicate the core and pan SVs,
respectively of the cattle pangenome.

BosMut3.0 yak reference genome or at least one other genome (Fig. 2c, Supplementary Fig. 3a, b, Supplementary Data 7). These SVs were either biallelic (insertions, deletions and divergent alleles) or multiallelic (40.8%, spanning 76 Mb) with more than one non-reference path (Fig. 2c), and tended to be enriched in repetitive DNA regions (Supplementary Fig. 3c). For genes present in the reference genome, we found that 97,258 SVs (33.1% of all SVs) are in potential expression regulatory regions (including the introns and the ± 5 kb flanking regions of a gene, as analyzed in this study) or coding sequence (CDS) in the bovine species. Among these SVs, we found 95,465 (98.2%) in potential expression regulatory regions and only 1793 (1.8%) in coding sequences (Fig. 2c, Supplementary Fig. 3a, Supplementary Data 7). We also discovered 38,090 multiallelic SVs (12.97% of all SVs) influencing 54.4% of the reference genes (Fig. 2c, Supplementary Fig. 3a, Supplementary Data 7). These results indicate a significant effect of SVs on the integrity and expression of genes.

Next, we used the graph-genotyping software Vg (v1.36.0)[26] on 386 bovines (233 yaks, 140 cattle, 4 bison, 8 wisent and one gaur, including the novel genome sequences for assembling the super-pangenome) for which resequencing data with >6× coverage were available (Supplementary Data 1, 2)[2,3,8,20,21,27–36]. This yielded 610,921 genotyped SVs, from which 57,432 were retained after quality filtering (Fig. 2d). The accuracy of SV detection was demonstrated by a high recall rate (0.96) and differences of SV-deletion genotype depth (Supplementary Fig. 3d). The number of genotyped SVs approached a plateau as more samples were added (Fig. 2e).

An examination of the global genetic structure of yaks by means of principal component analysis (PCA), Admixture, TreeMix and D-statistic analyses of SVs across samples indicated that most SVs are present within species or groups of related species, but with some exchange of haplotypes between yak and taurine cattle (Supplementary Figs. 4, 5). In 30.6% of SVs we observed strong linkage disequilibrium (LD, $R^2 > = 0.45$) with adjacent (within 50 kb) SNPs (Supplementary Fig. 3e).

## SVs specific to wild yaks are associated with high-altitude adaptation

We calculated the fixation index ($F_{ST}$) between the wild yak group and closely related low-altitude bovines (bison, wisents and European taurine) and used an outlier approach ($F_{ST} = 1$) to identify genomic regions that have potentially undergone selective sweeps between these two groups according to previous studies for such the closely related species[37–40]. There are limitations to applying this approach to compare between species, as it cannot exclude other causes of variation in gene flow; however, it can identify candidate loci undergoing selection. We further examined $\Phi_{ST}$ that was designed to mainly examine between-group divergence. Both showed a high consistence (Fig. 3a). Domestic yaks and Asian taurine were not included in this analysis because of their introgression[3] (Supplementary Figs. 4, 5). We detected 4830 SVs overlapping with 1051 genes having annotations enriched with HIF-1 signalling pathway hypoxia response, angiogenesis, cellular response to UV and DNA repair (Fig. 3a, b, Supplementary Data 8). These genes include *EPAS1* (*HIF2a*) of the HIF (hypoxia inducible factor) pathway, identified previously as conferring hypoxic adaptation in humans, horses, dogs and snakes[41–44]. A 254 bp insertion in the last intron of *EPAS1* in wild yaks overlapped a LINE1 element (Fig. 3c, Supplementary Fig. 6a). Both LINE1 and *EPAS1* DNA methylation have been shown to be involved in an epigenetic high-altitude adaptation of Andean humans[45]. This SV variant was fixed across all wild yaks, but had a lower frequency in domestic yaks (Fig. 3g) indicating that it is not crucial for their survival. The insertion decreased the strength of the promoter in a luciferase assay, possibly explaining why expression of *EPAS1* is lower in wild yaks than in cattle (Fig. 3e). Similar down-regulation of *EPAS1* transcription occurs in Tibetan humans[46].

We further observed that a 155 bp deletion (MB-hap-1) in both wild and domestic yaks occurred in the second intron region (Fig. 3d, h, Supplementary Fig. 6b) of *MB*, which encodes the oxygen storage protein myoglobin (MB) expressed in striated (heart and skeletal) muscles of vertebrates[47,48] and which modulates the expression of hypoxia marker genes in cancer cells[49]. This region of the gene contains transcription factor binding sites for *EP300* and *HIF-1α*. An in vitro reporter assay experiment indicated that the SV deletion variant of wild yaks has a higher enhancer activity than the insertion variant (Fig. 3f), which may at least partially explain the increased expression of *MB* as a plausible adaptation to hypoxia in both wild and domestic yaks. We identified other SVs in several other candidate high-altitude adaptation genes (Supplementary Data 8), including *PPARA* encoding a transcription factor associated with fatty liver disease and lipid metabolism disorder, the apoptosis regulator gene *BCL2*, the epidermal growth factor receptor gene *EGFR*, the insulin-like growth factor gene *IGF1R* which in dogs and mice is found to have an effect on body size, and the cytokine interleukin 6 receptor gene *IL6RA*. Further research may clarify the role of these genes in hypoxia adaptation.

Similar enrichment was found with SNPs. We detected 2259 SNPs with $F_{ST} = 1$, overlapping with 416 genes having annotations enriched with the HIF-1 signalling pathway, hypoxia response, and angiogenesis (Supplementary Data 9). Many key genes (*EPAS1*, *TEP1*, *EP300*, *TBX5*, *MMP2* and *PTEN*) have not been previously identified as such[1] and overlap with those identified via SVs. For example, we found an amino acid sequence variant (EPAS1.p.H733Y-yak) fixed in wild yaks and occurring more frequently in domestic yaks than in other species (Supplementary Fig. 7a). Western blot and qPCR of transfected HEK293T cells showed that Y733 induced a higher expression than H733 (Supplementary Fig. 7b, c) of the downstream *EPAS1*, encoding the hypoxia-associated protein erythropoietin (EPO)[50]. Thus, H733Y is a gain-of-function mutation that increases the transcriptional activity mediated by *EPAS1* in response to hypoxia. We observed strong LD ($R^2 = 0.64$) between the SV (254 bp) and SNP (H733Y) (Supplementary Fig. 7d), similar to that found for an SV and an SNP in Tibetan human *EPAS1* reported to be associated with reduced haemoglobin concentration[51].

## SVs in domestic yaks obtained from cattle and the origin of white yaks

To further investigate the origin of SVs in domestic yaks, we used SNPs with strong LD in the 50-kb segments flanking phased SVs from 30 domestic yaks and QTP cattle to construct a Neighbour-Joining (NJ) tree and trace the species origin of the SV haplotype. In this way, we identified 26,591 SVs (46.3% of all SVs identified via resequencing) involved in genetic exchange between yaks and QTP cattle (Supplementary Fig. 8a, b). From these SVs, 91.5% overlap with transposon elements (Supplementary Data 10). Interestingly, only 11.6% of yaks, mainly centered in the northern QTP near the wild yaks' population, did not harbour cattle SV haplotypes, while the level of cattle SV introgression decreased with longitude from east to west (Supplementary Fig. 9a, b). As expected, no yak SV introgression was found in cattle from either Europe or North America (Supplementary Fig. 9c). In the phylogeny of 47 genomes, individuals with a high degree of SV introgression were separated from monophyletic groups of relatively pure yak or cattle (Supplementary Fig. 9d).

Haplotypes of 11,486 SVs (20.0% of all SVs) overlapped with 1151 genes introgressed from domestic yaks to QTP cattle (Supplementary Fig. 8a, b). Several of these genes were annotated to be associated with high-altitude adaptation (Supplementary Fig. 10b, Supplementary Data 11). In contrast, 8557 SVs (14.9%) overlapping with 686 genes introgressed from cattle to domestic yaks appear to be involved in disease resistance, morphogenesis, embryonic skeletal system development and muscle fiber development (Supplementary Figs. 8a, b, 10d, Supplementary Data 12). Interestingly, 6547 SVs (11.4%) show

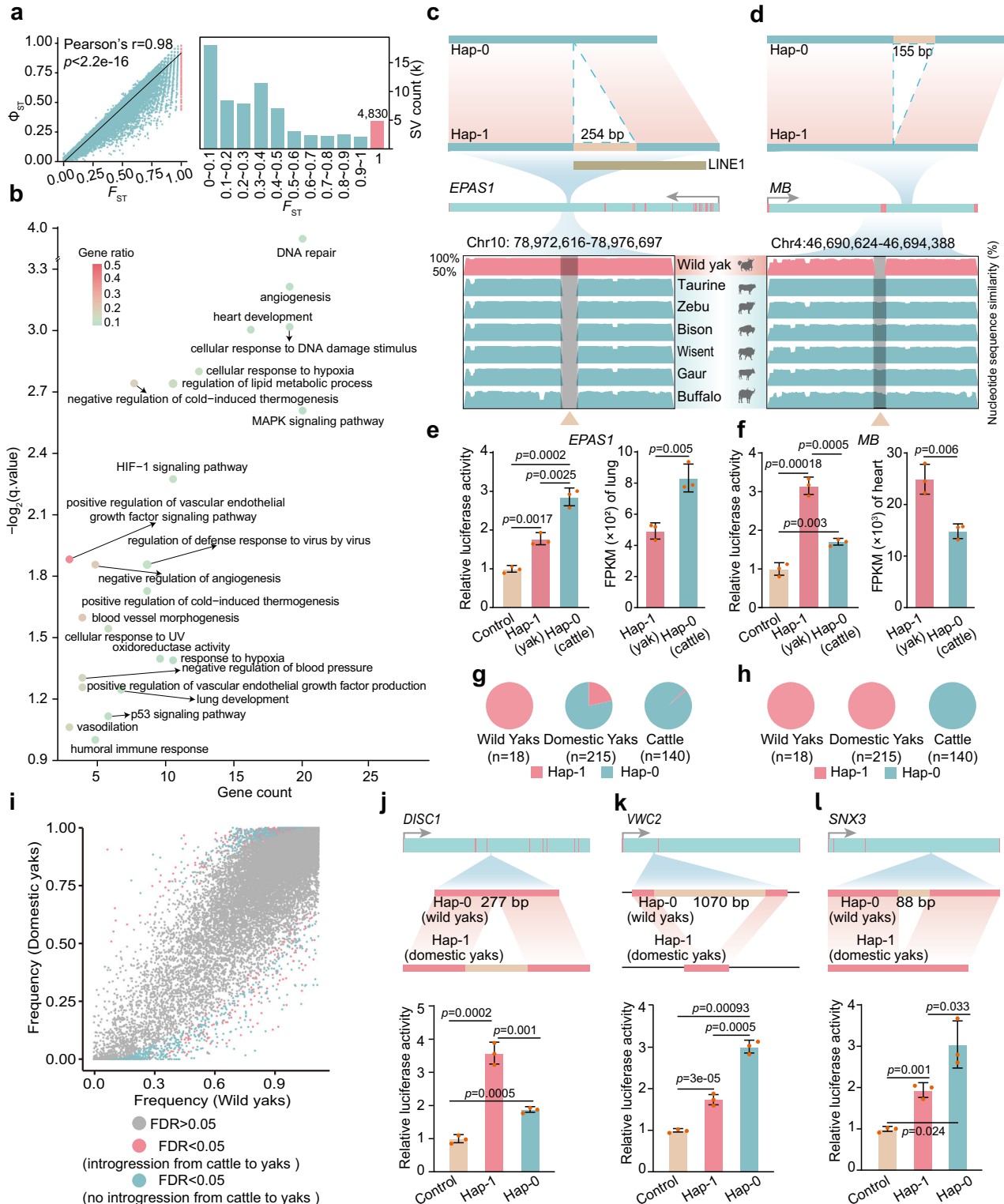

**Fig. 3 | SVs contributed to high-altitude adaptation and domestication of yaks.**
**a** $F_{ST}$ and $\Phi_{ST}$ between SV haplotypes in wild yaks and other bovine species (wisent, bison, European taurine). There is a significant linear relationship between the $F_{ST}$ and $\Phi_{ST}$ (Pearson's $r = 0.98$, $p < 2.2e-16$). **b** GO and KEGG enrichment analysis of the candidate genes of the identified SVs. **c, d** The two SV-haplotypes of *EPAS1* (facultative adaptation) and *MB* (obligate adaptation) genes are determined by the presence or absence of 254 bp and 155 bp in the intron, respectively. Hap-0: SV haplotypes in non-QTP cattle, guar, bison, wisent and outgroup buffalo. Hap-1: SV haplotype in wild yaks. A retrotransposons LINE1 that overlaps with SV of *EPAS1* is indicated with brown bar. VISTA plot of conserved shows sequence similarity of non-coding elements (CNEs) around and in the SVs of *EPAS1* and *MB* (indicated by triangles) to the BosMut3.0 reference. **e, f** Luciferase signals and expression of two SV-haplotypes of *EPAS1* and *MB*. Values are shown as means ± SD from three biological replicates; Exact $p$-values are shown; two-sided Student's $t$-test. **g, h** Frequencies of SV-haplotypes of *EPAS1* and *MB* in domestic yak, wild yak and cattle from graph genomes. **i** Frequency distribution of SVs haplotypes (dots) in wild and domestic yaks. **j–l** SV-haplotypes (Hap-0, mainly in wild yaks; Hap-1, mainly in domestic yaks) of SV-*DISC1*, *VWC2* and *SNX3* and corresponding luciferase signals. Values are shown as means ± SD from three biological replicates; Exact $p$-values are shown; two-sided Student's $t$-test. Source data are provided as a Source Data file.

bidirectional introgression (Supplementary Figs. 8a, b, 10d, Supplementary Data 13). For example, the yak-specific *EPAS1* SV-haplotype is present in high-altitude Tibetan cattle on the QTP (Supplementary Fig. 8c, d), whereas the cattle *EPAS1* haplotype occurs in low-altitude domestic yaks.

Two serial translocations of *KIT* in colour-sided cattle have produced two new alleles, Cs6 (on chromosome 6), and Cs29 (translocated to chromosome 29), within *LUZP2* (Supplementary Fig. 11a)[52]. Introgression of these alleles into yak has been proposed to have resulted in the occurrence of the white coat colour trait in yaks[52,53]. A GWAS of the association of SNPs or SVs with coat colour, as well as contrasting the SNPs of white and black yaks via the $F_{ST}$ and XP-CLR, indicate peaks on chromosome 6 and 25 (Fig. 4a), which coincide with those identified for colour-sided cattle[52]. We further found that the cattle-Cs6 allele has been introgressed into the peak region of chromosome 6 of colour-sided yak (Hap2-Cs6, Fig. 4e–h). For white yak, a primary contig assembly and two sets of completely phased contig assemblies from HiFi reads revealed a Hap1-SCs6 haplotype consisting of three copies of the same segment present in Hap2-Cs6 from colour-sided cattle (Fig. 4h). This was confirmed by the phylogenies of segments (Fig. 4e–g). However, their relative coverage in resequencing data identified 9 genotypes for white yaks and 8 genotypes for colour-sided yaks (Supplementary Fig. 11b–d). Both colour variants on chromosome 25 contain the yak Wt25 and/or Cs25 due to cattle Cs29 introgression.

SVs can modify the 3D genome by disrupting higher-order chromatin organization such as topologically associating domains (TAD)[54]. To gain insight into the regulation of upstream *KIT* gene expression by SV haplotypes, we generated Hi-C data from white and black yaks to map the interactions of GWAS, $F_{ST}$, and XP-CLR overlapping peak regions. We found that the topologically associating domains (TAD) region in white yak is a closed (repressed) B-compartment, while in the black yak this TAD region is an open (active) A-compartment (Fig. 4b). The BC fragment is essential for colour determination, suggesting that it is a potential super-enhancer element (Fig. 4b). This was supported by a *KIT* expression analysis, in which transcriptome datasets for ear tissue showed a significantly lower expression of *KIT* mRNA in white yaks than in black yaks (Fig. 4c, d). Immunohistochemical staining of *KIT* mRNA in croup skin and ear tissue sections of white and black yaks showed expression in the bulb of the hair follicle in black, but not in white yaks (Supplementary Fig. 12a). Furthermore, deposits of melanin pigment near melanocytes in the skin and ears was only observed in black yaks (Supplementary Fig. 12b). These results suggest that colour-sided yaks emerged by introgression of translocated and rearranged SV alleles from colour-sided cattle and that a further rearrangement with a longer tandem repeat generated an allele that in white yaks suppresses the expression of *KIT*.

### Contribution of SVs to early yak domestication

Affiliative behaviour and the suppression of aggression are likely to be prime targets of selection in the successful domestication of animals[55–58]. Identification of genes controlling aggression, tameness and sociability in yaks is therefore of considerable interest to understanding how yaks were domesticated and might also shed light on the genetic control of similar traits in humans. We considered SVs having a large difference (false discovery rate <0.05) in allele frequencies (AFs) between wild and domestic yaks as potentially involved in domestication (Fig. 3i). In this way, we identified 2354 SVs overlapping 654 candidate genes for domestication. Only a few of these SVs were detected previously in an analysis of fewer samples[8]. To identify SVs introgressed from cattle[3], we constructed phylogenetic trees of haplotypes in the 50-kb segments flanking the SVs, extracted from high-quality phased genomes that were heterozygous for these SVs. After exclusion of 1100 haplotypes, which in the trees clustered with their cattle homologues, we retained 1254 non-introgressed SVs, which

overlapped with 334 candidate domestication genes (Fig. 3i). Functional annotations showed an enrichment of neuron development, migration and synapse (Supplementary Data 14). Although enriched functions are largely consistent with those detected previously based on SNPs[2], the identified genes and specific functions were mainly different. For example, a 277 bp insertion in an intron of the Disrupted-in-Schizophrenia-1 (*DISC1*) gene is frequent in domesticated yaks but rare in wild yaks (Fig. 3j). Reporter assays suggested this insertion (DISC1-Hap-1) induces a higher expression than DISC1-Hap-0 without insertion (Fig. 3j). A chromosomal translocation within the human *DISC1* is reported to be associated with major mental disorders, including depression, schizophrenia, and bipolar disorder[59–62], which may suggest a role of the yak homologue in mental processes.

We also identified two frequently fixed deletions of 1070 bp and 88 bp in introns of *VWC2* and *SNX3*, respectively (Fig. 3k, l), that are not present in domestic yaks. *VWC2* which encodes the protein Brorin, is predominantly expressed in neural tissues of mouse adults and embryos and promotes neurogenesis in neural precursor cells[63], while *SNX3* encodes a sorting nexin protein, which plays an important role in mammalian neural tube closure. *SNX3*$^{-/-}$ mouse embryos have cranial neural tube defects and craniofacial defects with reduced face size[64]. This is consistent with the reduction of craniofacial size in domestic yaks compared with wild yaks. Reporter vector experiments showed that SV deletions in *VWC2* and *SNX3* decreased expression level (Fig. 3k, l). Additional domestication-related genes mediated by SVs (Supplementary Data 14) such as *STAT3*, *IGF1R* and *MAP1B* have orthologs in all animals and humans and warrant future investigation.

Finally, we scanned for potential selection signatures in the SNP dataset. We calculated $F_{ST}$ between wild and domestic yaks and identified regions with $F_{ST} > 0.3$ as candidate locations of selective sweeps. Again, we excluded genes introgressed from cattle. This yielded 2257 SNPs spanning 328 candidate genes, which were annotated as being associated with brain and neuronal development (Supplementary Data 15). The majority of these genes, including *SCRIB*, *PLXNB1* and *TTLL1*, have not been identified in previous studies[2].

## Discussion

We combined 28 new long-read and/or high coverage genomes with previously published assemblies for the construction of yak and cattle pangenomes, plus a super-pangenome and graph-genome of 7 bovine species. A dataset for 386 individuals further revealed genetic variants of evolutionary significance within and between these species. Assemblies, annotations and variant datasets are accessible via the *Bovini*-pangenome database (http://bovpan.lzu.edu.cn).

We caution that our dataset largely focuses on QTP yaks and Chinese cattle, and that published datasets used were generated by different novel long-read technologies and assembly procedures, which may result in variable representation of telomeric, centromeric and interspersed repeats, plus uneven coverage and contig size. Thus, ONT assemblies generally were of lower quality than PacBio HiFi generated genome sequences and resulted in lower coverage of repetitive regions as noted previously[10]. Nonetheless, our analyses have identified many SVs linked to high-altitude adaptation in both wild and domestic yaks, and also distinguished domestication signals differentiating wild and domestic yaks from SVs introduced by introgression from cattle. For example, several candidate genes were detected for high-altitude adaptation and domestication[1–3] in yaks, including *EPAS1*, *MB*, *DISC1*, *VWC1* and *SNX3*, that have not been reported previously. *EPASI* and *MB* are involved in hypoxia adaptation, but only *EPASI* segments have been introgressed from yak to cattle and vice versa. Furthermore, we have shown that introgression from cattle of a translocation SV haplotype spanning the *KIT* gene and the generation of a yak-specific rearrangement resulted in the origin of the colour-sided and white yak, respectively although the genetic mechanism to evolve into the latter need further studies. Our results

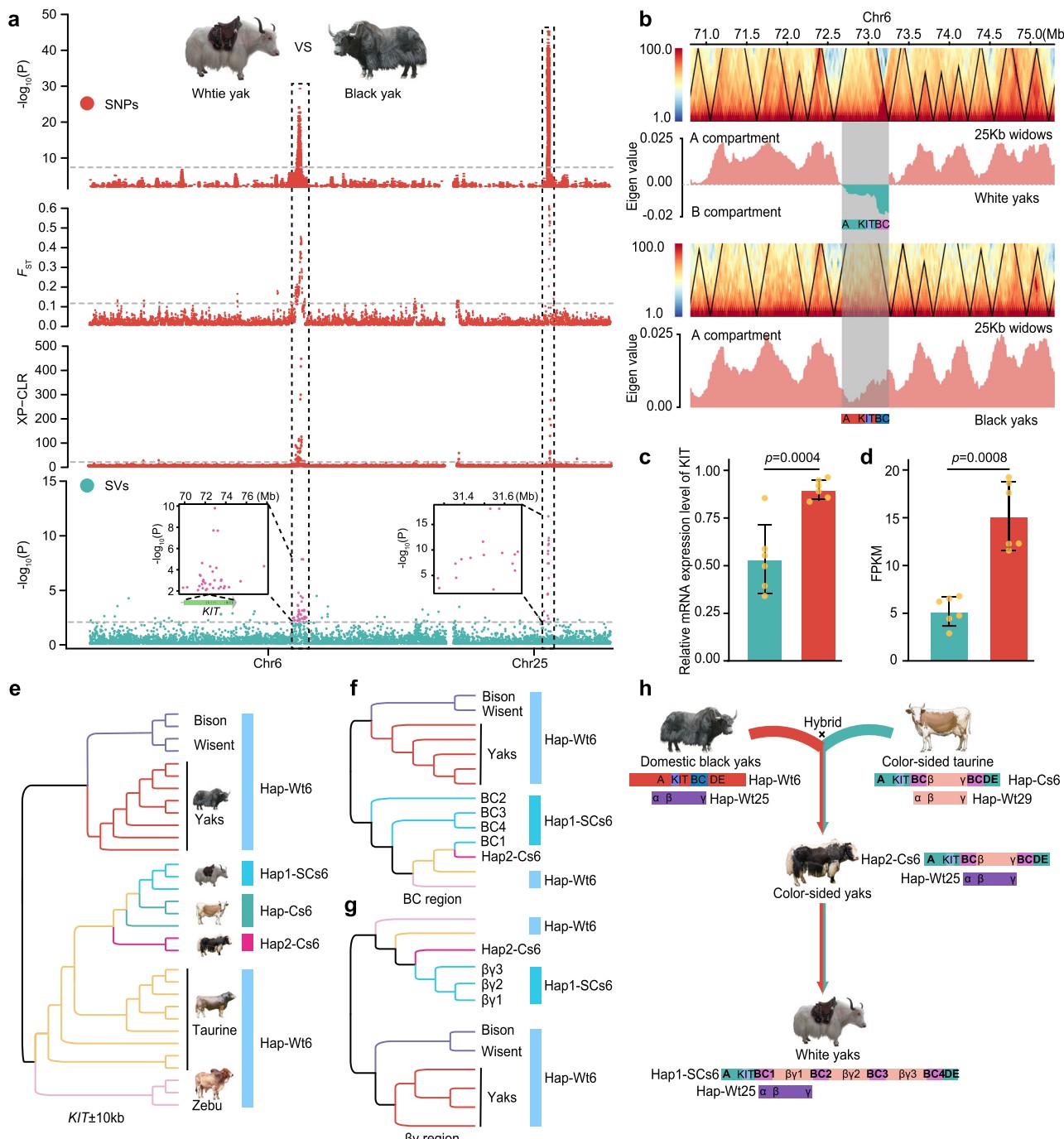

**Fig. 4 | The serial translocation SV with *KIT* introgressed from cattle contributed to white coat colours of yaks. a** Scans of chromosomes 6 and 25 of the association with coat colour and of $F_{ST}$ and XP-CLR values for black *vs* white yaks (red: SNPs; blue: SVs). **b** Chromatin interaction heatmap based on Hi-C sequencing data of white yak and BosMut3.0 (black phenotype), respectively, using the Bos-Mut3.0 genome as a reference. Black triangles indicate topologically associating domain (TAD). The BC fragment shows the strongest reciprocal signal within TAD. The bar graph block indicates the A/B compartment. Positive eigen values indicate A-compartment, corresponding to transcriptional activation regions. Negative eigen values indicate B-compartment, which correspond to closed chromatin regions. **c** Relative mRNA expression level and (**d**) FPKM of *KIT* in the ear tissue of black ($n = 6$) and white yaks ($n = 6$). Values are shown as means ± SD; Exact p-values are shown; two-sided Student's *t*-test. **e**–**g** The introgression between domestic yaks and colour-sided cattle with the NJ trees on SNPs of the phased *KIT* nearby ±10 kb, BC and βγ haplotype, respectively. **h** The most likely histories of the white and colour-sided yaks. Cs6: the allele on chromosome 6 of colour-sided yaks and cattle. SCs6: the allele on chromosome 6 of white yaks. Wt6/25: Wild-type allele on chromosome 6 or 25 of black and colour-sided yaks. Wt6/29: Wild-type allele on chromosome 6 or 29 of cattle. Chromosomes 6 and 25 of the yaks are homologous to chromosomes 6 and 29 of the cattle, respectively. Source data are provided as a Source Data file.

also indicate that nearly 90% of the sampled domestic yaks have been introgressed by cattle. Because these SVs affect breeding traits, it is therefore important to use them for guiding the genetic breeding of domestic yaks.

We identified candidate genes from both SNP and SV datasets and found the analysis of SVs was more effective in detecting genes that had been potentially subjected to natural and artificial selection. We further found that SVs not only modulate gene expression, but also may remove essential parts of a gene, implying that the gene repertoires of wild and domestic yaks are polymorphic. These observations support the hypothesis that SVs are a more important source of phenotypic variation than SNPs[8,31,65,66] and may also account for the hidden heritability exhibited by many traits[16]. However, the genetic bases of such traits may be also complicated by interspecific introgression. In addition to clarifying these genetic interactions, it will be interesting in future to record the effects of SVs on breeding values and to investigate if dedicated SV assays improve genomic selection in yaks and other livestock species.

## Methods

### DNA isolation and genomic sequencing

The wild yaks were collected from Hoh Xil Nature Reserve, Qinghai, China and Chang Tang Nature Reserve, Tibet, China, at the age of 3–5 years mainly from the naturally dead ones between 2003 - 2021. The domestic yaks and high-altitude cattle are collected from Tibetan herders' homes in the core distribution area. The Wannan cattle were sampled from Wanan cattle breeding farm, Anhui, China. The Brahman sample was a male individual of ~3 years old collected in 1998 in livestock breeding farm in Yunnan Province, China. All animal samples were collected legally in accordance with the Animal Care and Use Ethics policy. Animal collection and utility protocols were approved by the Animal Ethics Committee of College of Ecology, Lanzhou University. Genomic DNA of yaks and cattle was extracted from whole blood, muscle tissue and seminal fluid. For each HiFi library, an average insert size of ~15 kb was generated using the SMRTbell Express Template Prep Kit 2.0 (PacBio) and fractionated on the SageELF (Sage Science, Beverly, MA) into narrow library fractions according to the manufacturer's instructions. The constructed libraries were sequenced on three SMRT cells on a Sequel II instrument using Sequel II platform (Pacbio), in the circular consensus sequencing (CCS) mode. Raw data were processed using the CCS algorithm (v6.0.0) to generate highly accurate HiFi reads. For each Nanopore library, genomic DNA was size-selected ( > 20 kb) with a Blue Pippin (Sage Science, Beverly, MA) and processed using a ligation sequencing 1D kit (SQK-LSK108, ONT, UK) according to the manufacturer's instructions. Libraries were loaded on PromethION flow cells and sequenced according to standard protocols. To generate chromosome-scale genomes, the same individuals sequenced by HiFi were in turn used to construct Hi-C libraries, and a total of five constructed Hi-C libraries (DYXZ92, WY11, WY18, WNC04, Brahman01, Supplementary Data 1, 2) were sequenced on the BGI-Shenzhen MGISEQ-2000 platform with read lengths of 150 bp paired ends. For short-read sequencing of yaks and cattle, paired-end short insert (150 bp) libraries were constructed and sequenced on the BGI-Shenzhen MGISEQ-2000 platform of BGI (Shenzhen, China) or HiSeq 4000 platform of Illumina.

### RNA collection and sequencing

For transcriptome, immunohistochemistry and staining, we selected six healthy white yaks that were ~18 months old from the Bairi Tibetan Autonomous County, Gansu, China. We also selected six healthy black yaks, which were also ~18 months old, from a farm located in the Xiahe County, Gansu, China. We collected croup skin and ear tissue samples from these 12 yaks with a minimally invasive procedure for RNA extraction and histological section. The samples were trimmed and cut into small pieces, cleaned with RNase-free and DNase-free water and

immediately frozen in liquid nitrogen for storage until RNA isolation. Total RNA from each tissue sample was extracted by using TRIzol reagent (Thermo Fisher Scientific, CA, USA) according to the manufacturer's instructions. Next, six ear tissue replicates each from black and white yaks were selected for sequencing. Sequencing libraries were constructed using the NEBNext UltraTM RNA Library Prep Kit (NEB, Ipswich, MA) following the manufacture's introductions. The library preparations were sequenced using the Illumina HiSeq X Ten system. We also collected RNA samples from eight tissues (lung, brain, heart, liver, kidney, muscle, spleen, and testis) of yaks and cattle from public databases, with at least three replicates for each tissue (PRJCA000530, PRJNA177791, PRJNA573919, PRJNA362606; Supplementary Data 16). Combined with the downloaded RNA-seq data, we collected a total of 178 transcriptomes (Supplementary Data 16).

### Transcriptome analysis

After clipping adaptor sequences and removing low-quality reads, the clean reads from nine tissues of yaks and cattle were mapped to the respective reference genomes ARS-UCD1.2 and WY18 (Build Bos-Mut3.0) using Hisat2 (v2.1.0)[67] and gene expression levels were calculated in terms of FPKM (fragments per kilobase of exon model per million mapped fragments) by StringTie (v2.0.6)[68] with default parameters. To construct the inter-species gene expression matrix, we identified orthologous genes between yaks and cattle using BLASTp (v2.2.26)[69]. In total, we identified 19,710 1:1 orthologous gene. Differentially expressed genes that were significantly upregulated or downregulated between the two groups were identified by the Kolmogorov-Smirnov test using the R function ks.test with a threshold $P$-value of 0.01.

Differentially expressed genes between white and black yaks were identified with the DESeq2 package (v1.28.1) from Bioconductor. Genes were identified as differentially expressed if their expression level differed by two-fold.

### De novo assembly using HiFi and ONT reads

The long ( ~ 15 kb) and highly accurate ( > 99%) HiFi reads were assembled using hifiasm (v.0.14-r312)[70] with default parameters. Gfatools (v0.5-r234)[25] was used to convert sequence graphs in the GFA to FASTA format. For the ONT long-reads, we first used NextDenovo (v2.5.0, https://github.com/Nextomics/NextDenovo) for the self-correction. Next, we used wtdbg2 (v2.3)[71] with default parameters for yaks and cattle genome de novo assembly, after which contigs were polished with NextPolish (v1.01)[72] with three rounds of alignment with long reads and three rounds of alignment with short reads. In this study, in addition to own Hi-C anchoring own contigs, we selected one yak Hi-C (WY18, build BosMut3.0) and one cattle Hi-C (WNC04) to anchor all yaks and cattle to the chromosome level, respectively. The Hi-C short-reads were aligned to the scaffolds with Juicer (v1.5)[73], and the anchoring was performed with 3D-DNA (v180419)[74]. The Juicebox (v1.9.9)[73] assembly tools were then applied for the manual correction of the connections. The assembly completeness in genic regions was evaluated with Benchmarking Universal Single-Copy Orthologs (BUSCO, v3.0.2)[75] using the vertebrata_odb9 database.

### Annotation of genome sequences and phylogenetic analyses

For consistency in subsequent analyses, we annotated a total of 47 de novo genomes (22 yaks, 21 cattle, two bison, one wisent and one gaur) using a strategy combining de novo predictions, homology search and expression evidence. First, Augustus (v3.3.2)[76] trained by FGENESH[77] was used in de novo gene finding. Whole protein sequences from cattle, bison, wisent, water buffalo, goat, sheep, mouse, rat and humans were used to perform homologous gene structure predictions. The homology-based pipeline started with homology searching against a non-redundant collection of protein sequences using TBLASTN (v2.2.26)[69] with an E-value cutoff of 1E-5, followed by

selection of the most similar proteins for each region with homologous protein matching. Regions with homologous blocks <25% of query proteins were then excluded, and Genewise (v2.4.1)[78] was used to generate gene structures based on the homology alignments. RNA-seq reads were mapped to the genome by Hisat2 (v2.1.0)[67], then StringTie (v2.0.6)[68] was used to assemble transcripts, and the assembled transcripts were used to predict ORFs. Next, MAKER (v3.01.04)[79] was used to combine all these gene predications to construct the main structure of protein coding genes, and PASA (v2.3.3)[80] was used to predict alternative splicing types. Functions were assigned to the genes based on the best alignments to entries in the Swiss-Prot database (v15.10) using BLASTp (v2.2.26)[69]. Then we searched InterPro databases (v29.0) including the Pfam, PRINTS, PROSITE, ProDom, Superfamily and SMART databases to identify motifs and domains of the genes. GO terms for each gene were obtained from the corresponding InterPro entries. All genes were aligned against the KEGG database using the KEGG Automatic Annotation Server to identify pathways in which the genes might be involved. We annotated average 24,368 protein-coding genes, most of which could be aligned to entries in Swiss-Prot, InterPro or KEGG databases, and thus assigned putative functions.

In order to show more clearly the phylogenetic relationships of species in the *Bos genus*, we identified whole-genome SNPs using the show-snps (-ClrT) module of the MUMmer4 (v4.0.0rc1)[81] software across all 47 de novo genomes and the buffalo genome as phylogenetic outgroup. For quality control filtering, we removed SNPs with call rates <90% or with minor allele frequencies <0.05. A phylogenetic tree was constructed using Phylip (v3.698, https://evolution.genetics. washington.edu/phylip/install.html).

## Gene family-based pan-genome construction

For each yak de novo, a gene containing CDS with 100% similarity to other genes was removed by using the cd-hit-est of CD-HIT toolkit (v4.6)[82]. Protein sequences of the remaining genes were subjected to homologous searching by BLASTp (v2.2.26)[69]. OrthoMCL (v.2.0.9)[83] was used to deal with the BLAST result with the parameters of percentMatchCutoff = 50 and -I 1.5 for gene family clustering. We classified those clusters as three categories: core gene family clusters that were shared in all the yak individuals; near-core gene family clusters, which were missing one or two samples in our collection; and variable gene family clusters defined as those missing at least one samples. The gene family-based pan-genome for cattle and super-pangenome for the 7 bovine species were constructed as for the yak pan-genome.

## SNP and indel calling using short reads

In total, we collected 386 high-depth (>6×) short-read datasets, including 18 wild yaks, 215 domestic yaks, 140 cattle, 8 wisents and four bison and 1 gaur (Supplementary Data 1). Clean reads were aligned to the reference genome (BosMut3.0) using the BWA (0.7.12-r1039, http://bio-bwa.sourceforge.net/) with the default parameters. PCR duplicates were removed with the MarkDuplicates module in the Picard tools (v1.87, https://broadinstitute.github.io/picard/) and SNP and Indel calling was performed using the Genome Analysis Toolkit (GATK, v3.8, https://software.broadinstitute.org/gatk/). Finally, we identified 25,713,399 high-resolution SNPs (Fig. 2d).

## Multi-assembly graph genome construction and SVs calling

The minigraph (v0.15-r426)[25] with option '--inv no -xggs -L 10' was used to integrate 47 (22 yaks, 21 cattle, two bison, one wisent and one gaur) genome assemblies into a multi-assembly graph. The reference genome BosMut3.0 was used as the backbone of the multi-assembly graph and the remaining 46 assemblies were added successively. The bubble popping algorithm of gfatools (v0.5-r234)[25] was used to derive the structural variants from the multi-assembly graph for assemblies. Each bubble represents a structural variation, encompassing the start and end nodes of reference sequences as well as path traversing the start and end nodes. Structural variants were classified as biallelic if two paths were observed in a bubble and multiallelic if a bubble contained more than two paths.

We genotyped SVs in 386 high-depth (>6×) samples (Supplementary Data 1) using Panpop (https://github.com/StarSkyZheng/ panpop), including the novel de novo genome individuals for constructing graph-genome. PanPop was designed for merging, processing and filtering SVs with vg toolkit (v1.36.0)[26], which helped us to identify SVs more accurately. PanPop 'genotype' mode was used for SV calling with default parameters, which soft-filtered SVs of depth lower than one-third or higher than 3-fold of average depth. Finally, a set of high-confidence SVs, were obtained by removing genotypes with a call rate <0.7 or an MAF < 0.05. PCA and admixture analysis was performed with population-scale SVs using GCTA (v1.94.1)[84] and ADMIXTURE (v1.3.0)[85], respectively. With water buffalo as outgroup, a D-statistic test was performed for wild yaks, domestic yaks and QTP cattle, using the Dtrios programme within Dsuite (v0.4r28)[86] with the '-c' parameter based on whole-genome SVs. The $F_{ST}$ and $\Phi_{ST}$ for genome-wide SVs between the wild yak group and closely related low-altitude bovines (bison, wisents and European taurine) were calculated using VCFtools (v4.2)[87] and Arlequin (v3.5)[88], respectively.

## Determination of the introgressed SVs between cattle and domestic yaks

SVs introgressed between domestic yaks and cattle were firstly determined by the distributions of the resulting alleles in the co-occurring and allopatric regions. For each autosomal SV, we phased the SNP data using BEAGLE (v4.1)[89], selected the SNPs in the 50-kb haplotypes flanking the SV that were in close LD with the SV in domestic yaks and Tibetan cattle individuals and constructed haplotype trees. Comparing these trees with the species tree, we determined introgression directions: (1) from yaks to cattle with the haplotype found in cattle nested within the wild yak clades; (2) from cattle to yaks with the haplotype found in low-altitude yak nested within the cattle clade; and (3) mutual introgression with both scenarios.

## Selection scan for white yaks

Contrasting wild yaks (black, $n = 18$) and white yaks ($n = 31$) we carried out a GWAS for the SNPs and SVs data sets using Fisher's exact test and the dominance gene effect model in the Plink software (v1.90)[90]. We also calculated with SNPs the $F_{ST}$ and the cross-population composite likelihood ratio test (XP-CLR, sliding window of 50 Kb with step size of 25 Kb) using the VCFtools (v4.2)[87] and xpclr (v1.1.2)[91], respectively. The results were visualized using the qqman R package v0.1.4 (https:// CRAN.R-project.org/package=qqman).

## The origin of *KIT* serial translocations in white yaks

To retrieve the origin of *KIT* serial translocations in white or colour-sided yaks, we first assembled a high-quality white yak genome from HiFi CCS reads using Hifiasm (v.0.14-r312)[70] with default parameters. Hifiasm yields one primary contig assembly and two sets of partially phased contig assemblies. We firstly determined the cattle homologous A, BC, DE, α, and βγ regions[52] on the white yak genome and the black high-quality BosMut3.0 reference genome and found them located on chromosome 6 of both yaks. Then, the resequencing reads of the black yaks, colour-sided yaks, white yaks, solid colour cattle and colour-sided cattle were mapped to the BosMut 3.0 reference genome with BWA (0.7.12-r1039)[71] with the default parameters. We counted the average depth of the A, BC, DE, α, and βγ regions for all individuals and inferred nine genotypes in white and eight genotypes in colour-sided yaks.

For identification of the species origin of the BC and βγ regions, we used the SNPs in the same regions of the phased haplotypes from 30 black yaks, 31 white yaks, 40 colour-sided yaks, 30 colour-sided cattle and 40 randomly selected Tibetan cattle individuals for construction of the gene trees.

### Dual-luciferase assay of conserved non-coding SVs

For the highly conserved SVs of *EPAS1*, *MB*, *DISC1*, *VWC2* and *SNX3*, we extracted sequences containing the entire SV and the 200 bp flanking sequences from the genomes of yak and cattle respectively, and inserted them into the pGL3-promoter luciferase reporter vector (Promega, USA) before checking by sequencing (TsingKe Biotech or BIORN Life Science Co, Ltd, Nanjing, China). The HEK293T and PC12 cell lines were purchased American Tissue Culture Collection (ATCC, Rockville, MD, USA). The cell line was authenticated by short tandem repeat analysis. The cells were maintained in DMEM (GIBCO, USA) supplemented with 10% FBS (GIBCO) and 1% Antibiotic-Antimycotic (GIBCO) at 37 °C, 5% $CO_2$. These constructs as well as the pRL-TK plasmid as the internal control were transfected into HEK293T (*EPAS1*, *MB*) or PC12 (other genes) cells. After 24 h, transcriptional activity was investigated by the Dual-Luciferase Reporter Assay System (Promega) using a Multimode microplate reader (Spark Tecan, Switzerland). Each experiment was independently performed three times. All data are expressed as means ± SD, and statistical differences between groups were determined by two-sided Student's *t*-test, and a *p*-value < 0.05 was considered as a significant difference.

**Functional assay of linked amino acid mutation in *EPAS1* with SV Western blotting.** The cDNA sequences of EPAS1 Y733 (yak allele) and EPAS1 H733 (cattle allele) were synthesized by Tsingke (Beijing, China) based on genome sequencing results, and subcloned into the pIR-ES2EGFP vector. Cells transfected with EPAS1 Y733 and EPAS1 H733 were incubated in RIPA lysis buffer supplemented with protease inhibitors and phosphatase inhibitors. Following a 20-min incubation step at 4 °C and cleaned by centrifuging at 4 °C (15 min, 12,000 × g), the supernatants were collected. The protein concentrations of the supernatants were measured using the BCA assay kit. A total of 40 µg of the protein samples were boiled in SDS polyacrylamide gel electrophoresis (PAGE) sample buffer for 5 min and separated by 12% SDS-PAGE followed by electro-transfer onto polyvinylidene fluoride (PVDF) membranes. After blocking with 5% BSA in Tris buffered saline containing 0.1% Tween-20 at room temperature for 1 h, the PVDF membranes were incubated overnight with primary antibodies at 4 °C. They were then washed three times using TBST before incubating with secondary antibodies for 1 h at room temperature. Membranes were then washed again three times with TBST and exposed to chemiluminescent reagents. Images were captured using an ImageQuant LAS 4000 instrument (GE HealthCare). Immunoblotting was performed using Rabbit EPO Antibody (ab226956) and Mouse β-Actin Antibody as primary antibodies, and horseradish peroxidase-labelled anti-rabbit or anti-mouse antibodies as secondary antibodies.

**qPCR quantification.** To investigate whether endogenous EPO transcriptional up-regulation differs between EPAS1 Y733 (yak allele) and EPAS1 H733 (low-elevation allele), plasmids with yak or cattle sequences were constructed and were overexpressed in HEK293T cells under normoxic or hypoxic (1% $O_2$) conditions. For mRNA extraction, $5 \times 10^4$ HEK293T cells were plated on six-well plates in triplicate. When the cells reached 50% confluence, plasmids (1 µg per well) were transfected to cells after 48 h using Lipofectamine 3000. Real-time qPCR was performed using SYBR Premix Ex Taq II with first-strand cDNA to evaluate *EPO* expression.

### RT-qPCR for the *KIT* gene

Total RNA was isolated from 12 ear tissues (6 black yaks and 6 all-white yaks) using the TRIzol method, with quality and integrity detected using a NanoDrop 2000 spectrophotometer (Thermo Scientific, Waltham, MA, USA) and 2% agarose gel electrophoresis. The quantified RNA was then reversely transcribed into cDNA using Hifair®III 1st Strand cDNA Synthesis Kit (Yeasen Biotech Co., Ltd, Shanghai, China). Quantitative Real-time PCR was performed using an ABI QuantStudio 5 Real-Time PCR Detection System (ABI, USA) with SYBR qPCR Master Mix (Vazyme, China) and primer sets. Primer sequences used for qRT-PCR analysis of *KIT* expression were as follows: forward primer 5′-GGCGACGAGATTAGGCTGTT-3′ and reverse primer 5′-TCCCGTA-CAAGGGAAGGTCA-3′. Each experiment was independently performed six times and the $2^{-\Delta\Delta CT}$ analysis was normalized relative to GAPDH.

### Histology and immunohistochemistry

Croup skin and ear tissue samples of black and all-white yaks stored in 4% paraformaldehyde were washed in phosphate buffered saline (PBS), dehydrated with a gradient of alcohol solutions, cleared with xylene and infiltrated with paraffin wax. Samples were embedded in paraffin and sectioned into 4-micrometer-thick tissue sections. Ear tissue section was stained with hematoxylin and eosin (H&E) and croup skin sections were stained with DAB (diaminobenzidine). A rabbit anti-c-kit antibody (Bioss, China, bs-10005R) was used for immunohistochemical staining of croup skin and ear tissue samples, diluted at 1:200, by following a protocol involving the removal of wax from sections, antigen retrieval with Tris-EDTA (pH 8.0) in a microwave oven, and quenching autofluorescence. Sections were incubated with 4′,6-diamidino-2-phenylindole (DAPI) to stain with DAB, and were imaged with a light microscope.

### Reporting summary

Further information on research design is available in the Nature Portfolio Reporting Summary linked to this article.

### Data availability

The raw data of the long-read sequencing and Hi-C generated in this study have been deposited in the National Genomics Data Center (NGDC, https://ngdc.cncb.ac.cn) under accession number PRJCA016039, PRJCA018373 and PRJCA018427. The whole-genome resequencing and transcriptome data have been deposited at the NCBI BioProject database (https://www.ncbi.nlm.nih.gov/bioproject) under accession number PRJNA950586 and PRJNA952504, respectively. The details of data mentioned above and other downloaded publicly available data used in this study are provided in Supplementary Data 1, 2 and 16. The genome assemblies and VCF files (SNP, SV) have been deposited in the China National GeneBank DataBase (CNGBdb, https://db.cngb.org) with BioProject ID CNP0004693. The genome annotations have been deposited in Figshare database (https://figshare.com/articles/dataset/Untitled_Item/23988774). The genome assemblies, annotations and VCF files are also available on our database (http://bovpan.lzu.edu.cn). Source data are provided with this paper.

### Code availability

All code and software sources used in our paper are listed in the Methods section with corresponding cite of references. Other customized codes we used have been deposited in Github under https://github.com/XinfenLiu/yak_pangenome[92] and archived at https://doi.org/10.5281/zenodo.8260184.

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

## Acknowledgements

This work was supported by the Second Tibetan Plateau Scientific Expedition and Research program (Grant no. 2019QZKK0502 to J.Q.L.), Agricultural Science and Technology Innovation Program (Grant no. 25-LZIHPS-01 to P.Y.), National Natural Science Founda-tion of China (Grant no. U22A20447 to B.W.L.), National Key Research and Development Programs (Grant no. 2021YFD1200901 to K.X.L.), National Youth Talent Support Program (to Q.Q.) and Lanzhou University University's "Double First-Class" Guided Project Team Building-Funding-Research Startup Fee and International

Collaboration 111 Programme (Grant no. BP0719040). All the computation works were supported by the Big Data Computing Platform for Western Ecological Environment and Regional Development, and the Supercomputing Center of Lanzhou University.

## Author contributions

J.L., P.Y., and M.K. conceived and designed the project and together supervised the research. W.L. and X.L. collected materials and prepared all materials for RNA-seq and genome sequencing. W.L. wrote all codes for genome analsyes. W.L. performed genome assemblies and annotations and finished all pan-genome analyses. Z.Z. performed graph-based genome construction and variation calling. X.L. and W.L. conducted the analysis of genomic variations related to complex traits. X.L., W.L. and X.W. designed experiments for the target genes. X.L., W.L., J.A.L., Z.Z., X.W., J.Y., B.L., Y.Y., Q.Q., H.L., K.L., C.L., X.G., X.M., R.J.A. interpreted the data analysis and validation. J. L., J.L. and X.L. wrote the manuscript. All authors read, revised and approved the final manuscript.

## Competing interests

The authors declare no competing interests.
