## [Peer Review File · Nature Communications]

REVIEWER COMMENTS

Reviewer #1 (Remarks to the Author):

This work reports 28 *denovo* genomes from five bovine species (mostly yak) which were analyzed together with published WGS data. The data and findings of the work are very interesting and novel as structural variants differences have been seldom analyzed to infer selection and hybridization signatures left in the genomes of domestic species by the process of environmental adaptation or human-guided crossings. In general, this MS read relatively well and is worthy of publication but not before moderate revision. Please find below my main suggestions and concerns:

1. The authors start by recognizing that genes related to multifactorial traits are of particular interest to understanding the genetic bases of similar traits in humans (see lines 176-177) but this is quickly ignored as all the paper got focused on mendelian or near mendelian traits (1-2 genes explain the variance of the entire trait). I do not appreciate this sort of long-shot aim in a work that *per se* has the potential to provide more interesting findings regarding the evolutionary history of domesticated animals. This statement just belittles their work as they could not produce a single piece of evidence on the genetic architecture of such complex traits, and therefore, can be interpreted as not providing enough "novelty" that deserves publication. This is even more anecdotal in the following statement, at line 191, "a 277 bp insertion in an intron of the Disrupted-in-Schizophrenia-1 (DISC1) gene is frequent in domesticated yaks but rare in wild yaks". What is the meaning of this finding? does it contribute to the message of the manuscript?

2. Line 45. The authors refer that they have produced 28 de novo genomes among which 7 were Asian cattle (4 taurine, 2 zebu, and 1 hybrid). I understand that the hybrid is between cattle. But why they did not rather sequence cattle x yak hybrids? it would be more interesting to understand the recombination patterns of interspecific crosses with yak as the cattle hybrids are well covered. This would help to understand whether post-mating selection or gametic incompatibility would play a role in the frequency of some SVs among yak x cattle crossbreds. As far as I know the yak x cattle hybrids are more limited in terms of altitude adaptation but do better than pure cattle. Further down, line 126, some conclusions were drawn regarding the absence of some SV haplotypes in other analyzed species that were fixed in wild yaks. But the uneven number of samples on which they based this conclusion precludes this kind of "hard" statement (i.e., 18 yaks, 2 bison, 7 wisents, 11 European cattle). How can be speculated that two bison genomes are enough to rule out the presence of an SV?

3. Judging by the title and abstract I was expecting more work and findings regarding major differences between domestic and wild yaks. Indeed, there is a growing body of uncertainty among the scientific community on whether or many pure wild yaks still exist. This work could offer a very good clarification of this issue if the authors focused on analyzing the SV differences between both. Often, these differences are pointed as the consequence of cattle introgression, and not just because of different selective pressures. I understand that finding differences between such close related taxa is way harder than finding differences between yak and cattle, but the findings would be also much more rewardable than the ones reported in this study, which by the way, are not very novel.

minor suggestions

refrain from abbreviating terms or words that are only mentioned once, e.g., line 178 large difference (FDR < 0.05) in allele frequencies (AFs).

Reviewer #2 (Remarks to the Author):

The authors investigate the evolutionary origins of yak traits linked to structural variants (SV) through multi-species phylogenetic and pangenome analysis. To achieve this they de novo assemble multiple wild and domestic yak as well as Asian cattle. They combine their data with publicly available assemblies from diverse members of the bovini tribe to build a multi-species pangenome. SVs were genotyped using short-read data from a larger panel of individuals. SV

associations and impacts were validated with orthogonal datasets.

Major comments:

The manuscript is well designed and thorough however the writing requires major revisions for clarity.

It is not clear if all of the data used in this manuscript is publicly accessible. The manuscript needs a data availability section and should provide clear links to the data used. For instance, supplemental tables 1 and 2 should have a column with assembly accessions, SRA/ERA or equivalent numbers, or a link to the data source instead of referencing manuscripts or simply stating "this study".

Specific comments:

Line 60: Supplementary Table 4 should indicate which assemblies were from HiFi data.

Line 62: "resulted in a panel of" A panel of what?

Line 65: What is the origin of these SNPs? Methods are unclear.

Line 66: typo, should be form not for.

Line 74: soft core has a connotation in English that might be distracting. I suggest replacing with near-core or something similar.

Line 95: What do you mean by "genotyped"?

Line 105: Why obviously?

Line 117: What is the description of the light line in figure 2e?

Line 219: Since this section introduces introgressed SVs, I suggest you move it before the previous section "Contribution of SVs to yak domestication" so that the reader has a better understanding of the introgressed SVs mentioned there.

Lines 243-258: I'm confused about the roll of Cs29. It doesn't appear to matter which allele of Cs29 is present.

Line 284: Link does not work.

Non-exhaustive examples of text to rephrase for clarity:

Lines 23-25

Lines 28-29

Lines 32-34

Lines 77-78

Lines 122-123

Lines 310-311

RESPONSE TO REVIEWERS' COMMENTS

Dear Reviewers

Thank you for your comments concerning our manuscript entitled “Evolutionary origin of structural variations in domestic yaks” (ID: NCOMMS-23-01736-T). All comments were valuable and very helpful for revising and improving our paper, and allowing us to indicate better the significance of our study. We have studied the comments carefully and have made several corrections, which hopefully meet with your approval. We highlight (in red font) the major revisions and indicate below the line numbers in the new version of manuscript. The main corrections to the paper and our responses to the reviewer's comments are as follows:

Reviewer #1 (Remarks to the Author):

This work reports 28 de novo genomes from five bovine species (mostly yak) which were analyzed together with published WGS data. The data and findings of the work are very interesting and novel as structural variants differences have been seldom analyzed to infer selection and hybridization signatures left in the genomes of domestic species by the process of environmental adaptation or human-guided crossings. In general, this MS read relatively well and is worthy of publication but not before moderate revision. Please find below my main suggestions and concerns:

1. The authors start by recognizing that genes related to multifactorial traits are of particular interest to understanding the genetic bases of similar traits in humans (see lines 176-177) but this is quickly ignored as all the paper got focused on mendelian or near mendelian traits (1-2 genes explain the variance of the entire trait). I do not appreciate this sort of long-shot aim in a work that per se has the potential to provide more interesting findings regarding the evolutionary history of domesticated animals. This statement just belittles their work as they could not produce a single piece of evidence on the genetic architecture of such complex traits, and therefore, can be interpreted as not providing enough "novelty" that deserves publication. This is even more anecdotal in the following statement, at line 191, "a 277 bp insertion in an intron

of the Disrupted-in-Schizophrenia-1 (*DISC1*) gene is frequent in domesticated yaks but rare in wild yaks". What is the meaning of this finding? does it contribute to the message of the manuscript?

[Response]: We realize that the previous lines 176-177 “Genes controlling aggression, tameness and sociability in domestic yaks are of particular interest and may shed light on the genetic control of similar traits in humans” may create expectations that are not fulfilled in the rest of the manuscript. However, we did not mention “multifactorial traits”. In order to prevent a misunderstanding, we now have removed any reference to the multifactorial or complex nature of the traits, because it is not essential for our message. In addition, we added an introductory sentence (Lines 234-238) and have moved this section on early yak domestication (in the previous version starting with the sentence on lines 176-177) and now it follows the section on SVs of domestic yaks from cattle and the origin of white yaks. *DISC1* is a marker gene for major psychiatric disorders in humans, and its mutations are associated with depression, schizophrenia, and bipolar disorder. Of course, it is entirely unclear if these disorders exist in other species than humans, but we did find that a 277 bp insertion in the *DISC1* intron is frequent in domestic yaks, but rare in wild yaks. Combining our results with those of previous studies, we propose that mutations in the introns of *DISC1* may have modulated behavior of yaks during the early stages of domestication. This has been clarified in the revised version (Line 250-267).

2. Line 45. The authors refer that they have produced 28 de novo genomes among which 7 were Asian cattle (4 taurine, 2 zebu, and 1 hybrid). I understand that the hybrid is between cattle. But why they did not rather sequence cattle x yak hybrids? it would be more interesting to understand the recombination patterns of interspecific crosses with yak as the cattle hybrids are well covered. This would help to understand whether post-mating selection or gametic incompatibility would play a role in the frequency of some SVs among yak x cattle crossbreds. As far as I know the yak x cattle hybrids are more limited in terms of altitude adaptation but do better than pure cattle. Further down, line 126, some conclusions were drawn regarding the absence of some SV haplotypes in

other analyzed species that were fixed in wild yaks. But the uneven number of samples on which they based this conclusion precludes this kind of "hard" statement (i.e., 18 yaks, 2 bison, 7 wisents, 11 European cattle). How can be speculated that two bison genomes are enough to rule out the presence of an SV?

[Response]: Thank you for this thoughtful comment. We agree that the recombination pattern of interspecific crosses and post-mating selection or gamete discordance are indeed very interesting. For example, we discovered that the whitening of the yak's coat is primarily caused by chromosome recombination resulting from genetic introgression from cattle. In addition, the yak has lower productivity than cattle. Crossing of yaks and cattle is popular in regions of intermediate altitude and results in a unique highly productive offspring, the male sterile *dzo* and the fertile female *dzomo*. However, the pattern of genomic recombination during interspecific hybridization is complex. For example, it is not clear if the sterility of the *dzo* is due to abnormal sex chromosome recombination. In subsequent studies we will follow-up on these valuable comments on the basis of long-range whole-genome sequences for the yak-cattle hybrids.

Indeed, the yak x cattle hybrids have a better altitude adaptation than pure cattle but are kept at lower altitudes than yaks. In this and previous studies, we have found that genes involved in the hypoxia response pathway (including *EPAS1*, *EGLN1*, *EGLN2* and *HIF3a*) have been introgressed from yak to Tibetan cattle, which may have facilitated the adaptation of yak to high altitude environments (<https://doi.org/10.1038/s41559-018-0562-y>; DOI: 10.1038/s41467-018-04737-0). Regarding the comments of reviewer: on the uneven number of samples, we agree that our formulation caused a misunderstanding. As clarified in the revised text, the absence of the haplotypes in two American bisons is supported by their absence in other lowland bovine species, including 7 European bisons (Lines 124-126).

3. Judging by the title and abstract I was expecting more work and findings regarding major differences between domestic and wild yaks. Indeed, there is a growing body of uncertainty among the scientific community on whether or many pure wild yaks still exist. This work could offer a very good clarification of this issue if the authors focused

on analyzing the SV differences between both. Often, these differences are pointed as the consequence of cattle introgression, and not just because of different selective pressures. I understand that finding differences between such close related taxa is way harder than finding differences between yak and cattle, but the findings would be also much more rewardable than the ones reported in this study, which by the way, are not very novel.

[Response]: Thank you for this most interesting comment. Animal conservationists have estimated that there are only about 15,000 wild yaks in the world. To investigate whether pure wild yaks still exist, we carried out additional Admixture, TreeMix and D-statistic analyses, which indicate that wild yaks did not receive genetic introgression from domestic yaks or QTP cattle (Supplementary Fig. 4, 5, Lines 115-118). This is also supported by previously reported SNP-based clustering (<https://doi.org/10.1038/ncomms10283>; <https://doi.org/10.1186/s12862-020-01702-8>). In addition, in Supplementary Fig. 8e, we did not find any genetic introgression between wild yaks and Tibetan plateau cattle. In fact, breeders often cross male wild yaks with domestic yaks to produce in order yak breeds. Wild yaks live in isolated areas and domestic yaks cannot approach them due to the presence of predators. In conclusion, we believe that pure wild yaks exist, although a larger sample of wild yaks is needed to verify this.

We agree that “Often, these differences between wild and domestic yak are pointed as the consequence of cattle introgression, and not just because of different selective pressures”. In fact, this has been investigated in this study by selection of SV loci with different frequencies in wild and domestic groups (Fig. 3i). This method was mostly used to identify variable loci that differed between wild ancestors and domestic breeds (<https://doi.org/10.1186/s13059-020-02169-y>; <https://doi.org/10.1038/s41467-022-33366-x>). Less than half of the selected SVs (1100 out of 2354) were introgressed from cattle and we considered the other 1254 as being associated with the domestication.

“I understand that finding differences between such close related taxa is way harder than finding differences between yak and cattle, ...”. Indeed, the domestication of the yak around 7000 years ago took place later than the domestication of several other

livestock species (10,000-8,000 yr BP). We identified only 1254 non-introgressed SVs between wild and domestic yak populations, which accounted for 0.02% of all SVs.

minor suggestions

refrain from abbreviating terms or words that are only mentioned once, e.g., line 178 large difference (FDR < 0.05) in allele frequencies (AFs).

[Response]: Thank you for this comment. We have corrected the error in the revised manuscript (Line 239).

Reviewer #2 (Remarks to the Author):

The authors investigate the evolutionary origins of yak traits linked to structural variants (SV) through multi-species phylogenetic and pangenome analysis. To achieve this they de novo assemble multiple wild and domestic yak as well as Asian cattle. They combine their data with publicly available assemblies from diverse members of the bovine tribe to build a multi-species pangenome. SVs were genotyped using short-read data from a larger panel of individuals. SV associations and impacts were validated with orthogonal datasets.

The manuscript is well designed and thorough however the writing requires major revisions for clarity.

1. It is not clear if all of the data used in this manuscript is publicly accessible. The manuscript needs a data availability section and should provide clear links to the data used. For instance, supplemental tables 1 and 2 should have a column with assembly accessions, SRA/ERA or equivalent numbers, or a link to the data source instead of referencing manuscripts or simply stating "this study".

[Response]: Thank you for your helpful comments. To facilitate access to our data, we have uploaded all of the raw data generated in this study to the CNCB (<https://ngdc.cnbc.ac.cn/databases>) and NCBI database and provide the access codes in the supplementary data 1, 2, and 16.

Specific comments:

Line 60: Supplementary Table 4 should indicate which assemblies were from HiFi data.

[Response]: Thanks for this suggestion. According to your suggestion, we have marked the assemblies from HiFi data in the corresponding supplementary data 4 and 5.

Line 62: "resulted in a panel of" A panel of what?

[Response]: Thanks for pointing out our mistake. We have corrected the error in the revised manuscript (Lines 56-59).

Line 65: What is the origin of these SNPs? Methods are unclear.

[Response]: Thanks for this comment. We have modified this part of the method to clarify this in the revised manuscript (Lines 412-417).

Line 66: typo, should be form not for.

[Response]: Thanks for pointing out our mistake, which we have corrected.

Line 74: soft core has a connotation in English that might be distracting. I suggest replacing with near-core or something similar.

[Response]: Thank you for your professional comment. According to your suggestion, we have replaced soft-core with near-core in the revised manuscript.

Line 95: What do you mean by "genotyped"?

[Response]: genotyping refers to the identification of the alleles, which we believe to be correct (Line 92).

Line 105: Why obviously?

[Response]: Thank you for this comment. The graph-based genome enabled us to detect complex variants (multiallelic SVs), including those in highly diverse regions. Furthermore, we found that these complex variants affect half of the genes. However, we have modified this sentence (Line 102).

Line 117: What is the description of the light line in figure 2e?

[Response]: Thanks for pointing out our mistake. We have corrected the error in figure 2e. The red and light-grey lines, which indicate the Core-SVs and Pan-SVs for the cattle.

Line 219: Since this section introduces introgressed SVs, I suggest you move it before the previous section "Contribution of SVs to yak domestication" so that the reader has a better understanding of the introgressed SVs mentioned there.

[Response]: Thank you for your comment. We have made correction according to the Reviewer's comments (Lines 173, 233).

Lines 243-258: I'm confused about the roll of Cs29. It doesn't appear to matter which allele of Cs29 is present.

[Response]: Thank you for your comment. We indeed focused on Cs6 and were unable to assemble the sequence of Cs29 of the white yak, for which we would need more samples.

Line 284: Link does not work.

[Response]: Thank you for your comment. We have updated the link in the revised manuscript, which can be accessed using Google Chrome (Line 285).

Non-exhaustive examples of text to rephrase for clarity:

Lines 23-25

Lines 28-29

Lines 32-34

Lines 77-78

Lines 122-123

Lines 310-311

[Response]: Thank you for your professional comment. We have rephrased the text in the revised manuscript (Lines 23-29, Lines 31-32, Lines 74-77, Lines 119-121, Lines 303-305) and also made several other rephrasings (red font).

REVIEWER COMMENTS

Reviewer #1 (Remarks to the Author):

I thank you for the amended version and your responses to my concerns about the original version of the MS. I read over the rebuttal files, and I have some reservations about using the F_{st} outlier method to identify selection signatures between species. The authors discuss the use of F_{st} outlier testing to detect positive selection in wild yaks and other bovine species on Lines 124-126. I'm not sure what they were comparing. The frequency of SV- or SNP-alleles? Anyway, the F_{st} outlier method is appropriated for intraspecific genetic data (between populations of the same species), not interspecific genetic data (between different species). This test does not take into account the allele's/SV's genealogy or the mutational evolutionary model. Given this, I feel the test findings will not accurately reflect the selection forces.

Reviewer #2 (Remarks to the Author):

Thanks to the authors for addressing the majority of my concerns. My only remaining major concern is the clarity around the color sidedness analysis (fig4). Once this has been addressed I would be happy to support publication.

Line 61: The response to comments states that you have marked the assemblies from HiFi data in Supplementary Data 4 and 5, but only Data 4 has been marked.

Lines 119-121: Sentence is still difficult to read. I suggest the following rearrangement "In 30.6% of SVs we observed strong linkage disequilibrium ($LD, R^2 \geq 0.45$) with adjacent (within 50kb) SNPs (Supplementary Fig. 3e)."

Line 120: define LD here instead of below at line 168.

Lines 198-199: Now I'm confused about the introduction of Chr25. Was the previous figure mislabeled? As far as I can tell the new and old figures 4a are identical but you have switched the label from chr29 to chr25. The F_{st} analysis says that Chr25 is important but figure 4h suggests that all yaks share hap-Wt25. Also, what is the relationship between cattle Hap-Wt29 and yak Hap-Wt25?

Line 283: Link still does not work.

RESPONSE TO REVIEWERS' COMMENTS

Reviewer #1 (Remarks to the Author):

Thank you for the amended version and your responses to my concerns about the original version of the MS. I read over the rebuttal files, and I have some reservations about using the F_{ST} outlier method to identify selection signatures between species. The authors discuss the use of F_{ST} outlier testing to detect positive selection in wild yaks and other bovine species on Lines 124-126. I'm not sure what they were comparing. The frequency of SV- or SNP-alleles? Anyway, the F_{ST} outlier method is appropriated for intraspecific genetic data (between populations of the same species), not interspecific genetic data (between different species). This test does not take into account the allele's/SV's genealogy or the mutational evolutionary model. Given this, I feel the test findings will not accurately reflect the selection forces.

[Response]: Thank you for this thoughtful comment. We agree that in general the F_{ST} outlier method is better suited for comparing genetic structure and selection pressure between different populations of the same species. Indeed, within species F_{ST} is low in regions that have not been subject to selection and selection increases the F_{ST} , while between species F_{ST} may be high throughout because of the divergence of species. However, we believe that this does not preclude the use of the F_{ST} outlier method for closely related species. Our Fig. 3a shows by far most F_{ST} of SVs between the bovine species are $\ll 1$, so it seems plausible to consider SVs with $F_{ST} = 1$ as candidates for regions subject to selection (Lines 123-127). Several previous papers have used the same strategy. For example:

- 1) With the Denisovans, Neanderthals and *Homo sapiens* (doi: 10.1371/journal.pgen.1006675, doi:10.1093/molbev/msv141, doi:10.1038/nature13408);
- 2) Within the tree genus *Populus* (doi: 10.1111/nph.16215);
- 3) Within the bird genus *Corvus* (doi: 10.1038/s41467-020-17195-4), *Paridae* (doi:

10.1093/molbev/msaa157);

4) Within the rodent genus *Spalax* (doi:10.1073/pnas.2018123117/-/DCSupplemental);

5) For the rodent zokor species (doi:10.1073/pnas.2121819119/-/DCSupplemental);

6) Within the butterfly genus *Heliconius* (DOI: 10.1126/science.ade000).

Of course, no strategy to detect selection signatures is considered perfect and always need additional evidence, such as functional effect of mutation.

Reviewer #2 (Remarks to the Author):

Thanks to the authors for addressing the majority of my concerns. My only remaining major concern is the clarity around the color sidedness analysis (fig4). Once this has been addressed I would be happy to support publication.

Line 61: The response to comments states that you have marked the assemblies from HiFi data in Supplementary Data 4 and 5, but only Data 4 has been marked.

[Response]: Thanks for pointing out our mistake. We have corrected the error in Supplementary Data 5.

Lines 119-121: Sentence is still difficult to read. I suggest the following rearrangement "In 30.6% of SVs we observed strong linkage disequilibrium (LD, $R^2 \geq 0.45$) with adjacent (within 50kb) SNPs (Supplementary Fig. 3e)."

[Response]: Thank you for your helpful comments. According to your suggestion, we have rearranged this description (Lines 119-120).

Line 120: define LD here instead of below at line 168.

[Response]: Thanks for pointing out our mistake. We have corrected the error in the revised manuscript (Line 168).

Lines 198-199: Now I'm confused about the introduction of Chr25. Was the previous

figure mislabeled? As far as I can tell the new and old figures 4a are identical but you have switched the label from chr29 to chr25. The Fst analysis says that Chr25 is important but figure 4h suggests that all yaks share hap-Wt25. Also, what is the relationship between cattle Hap-Wt29 and yak Hap-Wt25?

[Response]: Thank you for your most useful comment. Yes, in figure 4a of the previous manuscript, we mistakenly labeled Chr25 of yaks as Chr29. Therefore, we made the correction in the revised version. In the genome assembly of the yak, chromosomes 6 and 25 of the yak are homologues of chromosomes 6 and 29 of the cattle, respectively, differing only in chromosome numbering. We have carefully revised all chromosome numbering in the latest manuscript to resolve your confusion (Lines 198, 212, and Fig. 4).

By GWAS, F_{ST} and XP-CLR, we found that allelic mutations in Chr25 be involved coat colour changes in yaks. However, the white individual we used for genome sequencing had the same Chr25 sequence as the black yaks, whereas cattle Hap-Wt29 and yak Hap-Wt25 indicate the same genotypes as the pure color cattle and pure black yak, respectively (see Supplementary Fig. 11a). In addition, relative coverage, sequence assembly and chromatin interaction analyses support that allelic mutations in Chr6 (CS6 and SCs6) as the most likely cause the coat colour change in yaks. Therefore, Fig. 4h only shows a model of yak coat colour change due to allelic mutations in Chr6. The involvement of Chr25 need to be further explored on the basis of additional data. The Hap-Wt25 of the yaks are homologous haplotypes to Hap-Wt29 of the cattle.

Line 283: Link still does not work.

[Response]: Thank you for this comment. Due to recent network maintenance at our school, external access is currently unavailable. We are actively communicating to resolve this issue as soon as possible. You will be able to access it shortly.

REVIEWERS' COMMENTS

Reviewer #1 (Remarks to the Author):

I hold numerous concerns regarding the utilisation of elementary F-statistics (F_{st}) at the inter-species level. Despite their foundation in F-statistics, there already exist derived parameters that aim to incorporate the genealogy of alleles (SNPs), such as Φ_{ST} . These derived parameters represent a middle ground in terms of their application of statistical measures at the inter-species level. While I comprehend the line of argumentation put forth by the authors, which essentially rests upon the precedent principle, I am uncertain whether the respective distinctions, *mutatis mutandis*, have been thoroughly considered to sufficiently justify the use of simple Wright's F_{ST} in this particular case.

Regrettably, I am unable to offer any alternative suggestions at this juncture, and the editor is left to make a decision unaided.

Response to editor's and reviewers' comments

Dear editor and reviewers

Thank you for your letter and for the reviewers' comments concerning our manuscript entitled "Evolutionary origin of structural variations in domestic yaks" (ID: NCOMMS-23-01736-B). All comments were valuable and very helpful for revising and improving our paper, and allowing us to indicate better the significance of our study. We have studied the comments carefully and have made several corrections, which hopefully meet with your approval. The main corrections to the paper and our responses to the reviewer's comments are as follows:

Reviewer #1 (Remarks to the Author):

I hold numerous concerns regarding the utilisation of elementary F-statistics (F_{ST}) at the inter-species level. Despite their foundation in F-statistics, there already exist derived parameters that aim to incorporate the genealogy of alleles (SNPs), such as Φ_{ST} . These derived parameters represent a middle ground in terms of their application of statistical measures at the inter-species level. While I comprehend the line of argumentation put forth by the authors, which essentially rests upon the precedent principle, I am uncertain whether the respective distinctions, *mutatis mutandis*, have been thoroughly considered to sufficiently justify the use of simple Wright's F_{ST} in this particular case.

Regrettably, I am unable to offer any alternative suggestions at this juncture, and the editor is left to make a decision unaided.

[Response]: Thank you for this thoughtful comment. Wild yaks have long settled on the Qinghai-Tibet Plateau. Comparison of the wild yaks with other bovine species with allopatric distributions without gene flow is a good choice for identifying the adaptive genetic variations with selection signals. Generally, using the F_{ST} outlier method is more suitable for comparing the genetic structure and selection pressure among different populations of the same species. However, it seems reasonable to use it for

selection of candidate genes between closely related species in numerous case studies. (doi: 10.1111/nph.16215; 10.1038/s41467-020-17195-4; 10.1073/pnas.2018123117/-/DCSupplemental and 10.1126/science.ade00). However, it cannot exclude other causes of variation in gene flow although it can identify candidate loci undergoing selection. We added this caveat claim in the manuscript. Especially, following your suggestion, we also added Φ_{ST} calculations between species that also supported the F_{ST} results (see Fig.3a). Therefore, all of these results could convincingly suggest that the variants with $F_{ST} = 1$ should reflect the selection-related genetic signature.